# Tuning the apparent hydrogen binding energy to achieve high-performance Ni-based hydrogen oxidation reaction catalyst

Xingdong Wang[1,2,4], Xuerui Liu[1,4], Jinjie Fang[1,4], Houpeng Wang[2], Xianwei Liu[1], Haiyong Wang[1], Chengjin Chen[1], Yongsheng Wang[1], Xuejiang Zhang[1], Wei Zhu ®[1] & Zhongbin Zhuang ®[1,3] ✉

High-performance platinum-group-metal-free alkaline hydrogen oxidation reaction catalysts are essential for the hydroxide exchange membrane fuel cells, which generally require high Pt loadings on the anode. Herein, we report a highly active hydrogen oxidation reaction catalyst, NiCuCr, indicated by the hydroxide exchange membrane fuel cell with a high peak power density of 577 mW cm$^{-2}$ (18 times as high as the Ni/C anode) and a stability of more than 150 h (a degradation rate slower by 7 times than the Ni/C anode). The spectroscopies demonstrate that the alloy effect from Cu weakens the hydrogen binding, and the surface $Cr_2O_3$ species enhance the interfacial water binding. Both effects bring an optimized apparent hydrogen binding energy and thus lead to the high hydrogen oxidation reaction performance of NiCuCr. These results suggest that the apparent hydrogen binding energy determines the hydrogen oxidation reaction performance and that its tuning is beneficial toward high electrocatalytic performance.

The intensive use of fossil energy causes global warming. Hydrogen fuel cell is a technology that transforms the chemical energy stored in hydrogen directly into electricity with high efficiency, and its only emission is water. It acts as one of the practical and promising solutions to realize a carbon-neutral society. In the category of hydrogen fuel cells, hydroxide exchange membrane fuel cells (HEMFCs) developed rapidly recently[1–5]. Compared with the proton exchange membrane fuel cells, the HEMFCs may employ alternative platinum-group metal (PGM) free catalysts, less harsh bipolar plates, and cheaper membrane materials[6–10]. However, the successful commercialization and wide application of the HEMFCs are heavily dependent on the highly efficient hydrogen oxidation reaction (HOR) catalysts on the anode[8,11], because the PGMs (e.g., Pt, Ir, Pd, and Rh) have HOR performance in alkaline 2~3 orders of magnitude lower than those in acid[12,13]. As a result, much higher PGMs loading is required on the anode for HEMFCs, leading to high cost[2]. It is imperative to develop

highly active PGM-free anode HOR catalysts in order to achieve the ultimate target for HEMFC using PGM-free membrane electrode assembly (MEA)[14,15].

Nickel-based catalysts are the most promising candidates for PGM-free HOR catalysts in alkaline, but the HOR performance of pristine Ni is only 1/1000 to 1/100 of that of Pt[16]. Many efforts have been made to improve the HOR performance of the Ni-based materials. Alloying Ni with other transition metals[17,18], such as NiCu[19,20], NiCr[21], NiTi[22], NiAg[23], NiMo[24,25], NiW[26] and ternary alloy ($Ni_{5.2}WCu_{2.2}$[27], CoNiMo[28]) has been reported as an efficient approach to improve the activity and more than an order of the magnitude enhancement compared with pristine Ni has been achieved. Cooperating with non-metal elements, such as doping N or O in catalyst nanoparticle or substrate (e.g., $Ni_3N$[29], $Ni/Ni_3N$[30,31], $Ni_{ED}/XC-72$[32], $Ni(OH)_2-Ni/C$[33], $Ni@O_i$-Ni[34], Ni-400[35], Ni/N-CNT[36], and Ni/SC[37]), has been found beneficial for promoting the HOR performance as well. However, there are still HOR

[1]State Key Lab of Organic-Inorganic Composites and Beijing Advanced Innovation Center for Soft Matter Science and Engineering, Beijing University of Chemical Technology, Beijing 100029, China. [2]Research Institute of Petroleum Processing, SINOPEC, Beijing 100083, China. [3]Beijing Key Laboratory of Energy Environmental Catalysis, Beijing University of Chemical Technology, Beijing 100029, China. [4]These authors contributed equally: Xingdong Wang, Xuerui Liu, and Jinjie Fang. ✉e-mail: zhuangzb@mail.buct.edu.cn

performance gaps between those Ni-based catalysts and the state-of-the-art Pt-based catalyst[15]. Especially in the HEMFCs, the MEA using Ni-based anode only shows dozens to hundreds of mW cm$^{-2}$ peak power density with tens of hours stability[38]. Further improvement of the HOR performance for the Ni-based catalysts is desired but still challenging.

The hydrogen binding energy (HBE), which represents the binding strength of the HOR intermediates to the catalysts, is considered the primary activity descriptor for the HOR catalysts[39]. In alkaline conditions, the condition is more complex, and 2–3 orders of magnitude lower kinetics are found for Pt group metals[12,13]. It implies that the additional factors beyond HBE also control the HOR kinetics[40]. To uncover the factors that influenced the HOR kinetics, the researchers have paid intensive attention to the electrochemical double layer structure, where water, cations, and hydroxide ions exist. The adsorbed hydroxide is found to influence the HOR kinetics in alkaline significantly, and the bifunctional mechanism is proposed[41,42]. The interfacial water is important for the HOR kinetics as well. The catalyst surfaces have interfacial water with different reorganization energy[43,44] and connectivity of hydrogen-bond networks[45], leading to the altered HOR kinetics. For example, the interfacial water structure is controlled by the co-adsorbate cations (i.e., Li$^+$, Na$^+$, K$^+$, Rb$^+$, and Cs$^+$) on Pt surfaces, and the HOR activity can be enhanced using electrolytes with Li$^+$[46]. To count both influences from HBE and interfacial water, Yan and co-workers proposed the apparent HBE (HBE$_{app}$)[47,48], which is defined as HBE$_{app}$ = HBE − WBE, where WBE represents the Gibbs free energy of interfacial water adsorption and contributed to the HBE$_{app}$ negatively. The HBE$_{app}$ accounts for both bindings of hydrogen and water to the catalysts and is considered to be the activity descriptor for the HOR catalysts. However, to promote the HOR activity by controlling the HBE$_{app}$ of the catalysts is a lack of practice because it is difficult to design a single component catalyst with weakened interaction to one intermediate but strengthened for another[49].

Here, we fine-tune the HBE$_{app}$ of Ni by synthesizing heterostructured NiCuCr/C ternary catalyst to achieve a highly active HOR catalyst. Specifically, to optimize the intrinsic HBE, we select Cu as an

alloying agent to weaken the hydrogen adsorption of Ni because Cu has weak HBE and is miscible with Ni. To optimize WBE, a more hydrophilic element, Cr, in the form of oxide, is decorated on the surface of the catalyst to manipulate the solvation environment. The Cr$^{3+}$ is one of the ions with the lowest enthalpies of hydration, suggesting a strong binding to water[50]. Through this heterostructure design, optimized HBE$_{app}$ is achieved. The obtained NiCuCr/C catalyst demonstrates a high HOR activity in both RDE and MEA tests, with the mass activity of 72.6 A g$_M^{-1}$ at 50 mV in the RDE test and the peak power density of 577 mW cm$^{-2}$ at 80 °C in HEMFC, surpassing most of the reported PGM-free catalysts. A stability test of more than 150 h is displayed by the NiCuCr/C HEMFC, far beyond the durability of HEMFC using Ni/C as an anode. These results demonstrate the practice of tuning HBE$_{app}$ through the cooperation of selected elements and the heterostructure, and it has proved to be an efficient way to fabricate high-performance HOR catalysts.

## Results

### Morphological and physical characterizations

The NiCuCr/C catalysts were synthesized by impregnation. Nickel nitrate, copper nitrate, and chromium nitrate were used as the metal sources, and Ketjen ECP600 carbon was selected as carbon support. Citric acid and urea were added to regulate the particle sizes. The final NiCuCr/C catalyst was obtained by annealing at 470 °C in 5% H$_2$ and 95% Ar atmosphere. Three control samples (Ni/C, NiCu/C, and NiCr/C) were synthesized through similar processes by only adding the corresponding metal salts.

Figure 1a shows the transmission electron microscopy (TEM) image of the NiCuCr/C. The NiCuCr nanoparticles appeared with an average diameter of 16 nm (the bottom left insert of Fig. 1a shows the size distribution histogram) and were distributed uniformly on the carbon support. The selected area electron diffraction pattern (SAED, the upper right insert of Fig. 1a) demonstrated the good crystallinity of the nanoparticles, and the diffraction rings were assigned to the (111), (200), (220), and (311) facets of Ni alloy. The crystalline phase of

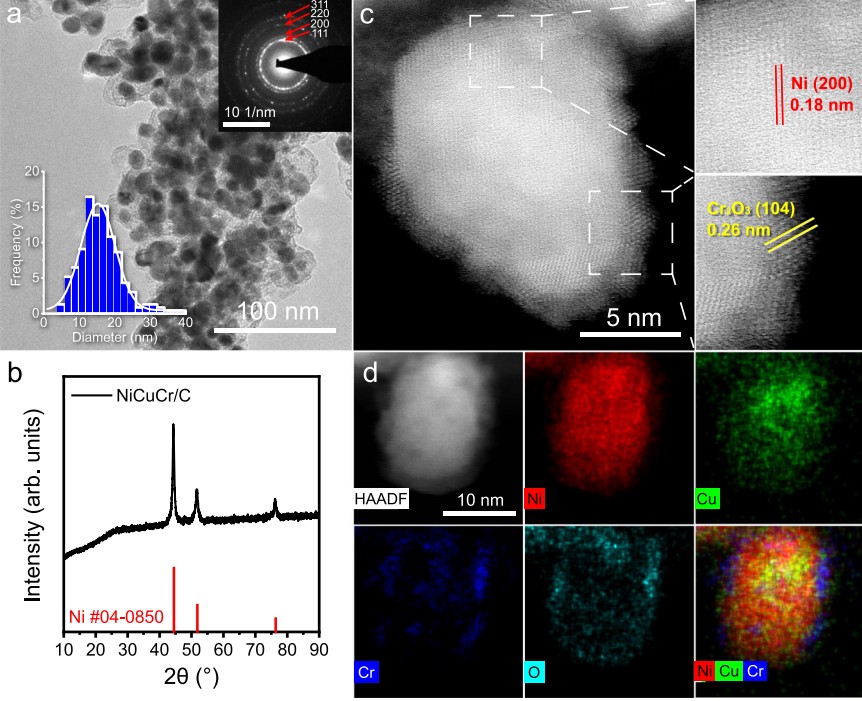

**Fig. 1 | Electron spectroscopy of the NiCuCr/C catalysts. a** TEM image of NiCuCr/C. The bottom left insert is a statistical histogram of the size of the particles. The upper right insert is the selected area electron diffraction pattern of NiCuCr/C. **b** The XRD pattern of NiCuCr/C. **c** HAADF-STEM image of a NiCuCr nanoparticle. Right enlarged areas show the lattice fringes. **d** HAADF-STEM elemental mapping image of a NiCuCr nanoparticle.

NiCuCr/C was further proved to be a Ni alloy by the XRD pattern (Fig. 1b). The shoulder at ca. 26° in the XRD pattern was contributed by the carbon support. High-angle annular dark-field scanning TEM (HAADF-STEM, Fig. 1c) and high-resolution TEM (HRTEM, Supplementary Fig. 1) image showed clear lattice fringes, and the lattice fringe with the spacing of 0.18 nm was assigned to the (200) facet of Ni. The nanoparticles displayed rough surfaces, and there were a lot of sub-nanometer-sized clusters existing on the surface of the nanoparticles. These clusters showed lattice fringes with a spacing of 0.26 nm, which were assigned to the (104) facet of $Cr_2O_3$. The elemental mapping images illustrated that Ni and Cu distributed homogeneously in the nanoparticle, while Cr was enriched at the edge of the nanoparticle, and the O distribution almost resembled that of Cr (Fig. 1d). It confirmed that the nanoparticles were composited with NiCu alloy, and Cr was phase-separated to the surface as clusters in the form of oxides. The metal contents of Ni, Cu, and Cr were measured by inductively coupled plasma-optical emission spectrometry (ICP-OES), which was 57.6, 3.9, and 1.1 wt%, respectively (Supplementary Table 1), and the rest was the carbon support.

Three control samples (i.e., Ni/C, NiCu/C, and NiCr/C) were synthesized using a similar process. Supplementary Fig. 2 shows the XRD pattern of the as-prepared Ni-based catalyst. The three main diffraction peaks at 44.5°, 51.8°, and 76.4° were similar to that for the Ni (JCPDS card No. 04-0850), demonstrating the formation of Ni-based catalysts. The TEM images (Supplementary Fig. 3a–c) show that the formed nanoparticles were well dispersed on the carbon substrate with a diameter of around 16 nm (Supplementary Fig. 3d–f). The average particle sizes were obtained from statistical analysis on TEM images and calculated from the XRD pattern using the Scherrer equation (summarized in Supplementary Fig. 4). The metal contents measured by ICP-OES were summarized in Supplementary Table 1, demonstrating the similarity of the control samples.

The chemical environments of the synthesized Ni-based catalysts were studied by X-ray photoelectron spectroscopy (XPS). Supplementary Fig. 5 shows the high-resolution Ni 2*p* XPS spectra. For Ni/C, the peak at 852.1 eV was assigned to the Ni(0) species, and the peaks at 853.6 and 855.2 eV were assigned to the Ni(II) species in NiO and Ni(OH)$_2$, respectively, coming from the surface oxidation. The Ni (0) peaks of NiCuCr/C, NiCu/C, and NiCr/C show a positive shift (-0.8 eV) compared with Ni/C, indicating the transferred electrons from Ni. Supplementary Fig. 6 shows the high-resolution Cu 2*p* spectra of NiCuCr/C and NiCu/C, indicating that Cu mainly existed in Cu (0) state. The NiCuCr/C illustrates a negative shift of -0.3 eV compared with the NiCu/C, demonstrating the tuning effect of Cr additives. Supplementary Fig. 7 shows the high-resolution Cr 2*p* spectra of NiCuCr/C and NiCr/C. They displayed two similar peak positions at 577.5 and 587.3 eV, indicating that the Cr was mostly in oxidized Cr(III) form. The C 1*s* XPS peak of the NiCuCr/C (Supplementary Fig. 8) can be deconvoluted into three peaks at 284.8, 286.1, and 289.5 eV, assigned to the C −C, C−O, and C=O bond, respectively.

The coordination environments of NiCuCr/C were further characterized by X-ray absorption fine structure (XAFS) spectra. Figure 2a shows the Ni K-edge X-ray absorption near-edge structure (XANES) spectra. The NiCuCr spectrum was close to that of Ni foil, demonstrating the metallic form of Ni in the NiCuCr. However, compared with Ni foil, NiCuCr went to a higher valence state with regard to the adsorption edge position, revealing the charge transfer out of Ni, which was consistent with the XPS results. Figure 2b shows the Fourier transform extended X-ray absorption fine structure (FT-EXAFS) and the oscillation spectra are shown in Supplementary Fig. 9a. The NiCuCr showed a major peak at 2.18 Å, which was assigned to Ni−Ni scattering referenced by the Ni foil. The NiO showed two peaks at 1.72 and 2.62 Å, which were assigned to Ni−O and Ni−Ni scattering, respectively. These peaks were not observed in the NiCuCr, demonstrating the mainly metallic form of Ni in NiCuCr. Wavelet transform EXAFS of NiCuCr

displayed similarities with respect to Ni foil (Fig. 2c, d). Figure 2e shows the Cu K-edge XANES spectra. Similar to that for Ni spectra, the NiCuCr spectrum was close to that of Cu foil, revealing the metallic form of Cu in the NiCuCr, which was consistent with the XPS result. Compared with Cu foil, Cu in the NiCuCr had a slightly higher valence. A major peak at 2.18 Å existed in the FT-EXAFS pattern (Fig. 2f, oscillation spectra are shown in Supplementary Fig. 9b), which was slightly smaller than the Cu−Cu scattering at 2.24 Å in Cu foil. This peak was assigned to the Cu −Ni scattering in CuNi alloy because of the smaller size of Ni than that of Cu. Only the Cu−Ni scattering was observed, demonstrating the good dispersion of Cu in the Ni host and the low Cu content. No scatterings from CuO (1.55 Å for Cu−O and 2.47 Å for Cu−Cu) were observed, demonstrating the metallic nature of Cu species in NiCuCr. It was further confirmed by the wavelet transform EXAFS (Fig. 2g, h). However, the coordination environment was much different for Cr. Figure 2i shows the Cr K-edge XANES spectra, and the NiCuCr spectrum was close to that of $Cr_2O_3$, which was consistent with the XPS result. In the FT-EXAFS pattern (Fig. 2j, oscillation spectra are shown in Supplementary Fig. 9c), the NiCuCr shows two major peaks at 1.60 and 2.55 Å similar to that for $Cr_2O_3$ foil, which was attributed to the Cr−O and Cr −Cr scatterings, respectively. While the Cr foil shows a major peak at 2.19 Å for the metallic Cr−Cr scattering, and it did not occur in the NiCuCr spectrum. The oxidized form of Cr in the NiCuCr was further confirmed by the wavelet transform EXAFS (Fig. 2k, l).

Based on the above characterization, the NiCuCr/C were uniform NiCuCr nanoparticles supported on carbon, and the NiCuCr nanoparticles had the structure of NiCu alloy particles with $Cr_2O_3$ clusters dispersed on the surface.

## Electrochemical performance

The HOR activity of the catalysts was evaluated by a standard three-electrode system with a rotating disk electrode (RDE) in $H_2$-saturated 0.1 M KOH. Figure 3a shows the polarization curve of the four synthesized Ni-based catalysts and the benchmark commercial 20% Pt/C. The steeply grown anodic current density with the potential implied the high HOR activity of the NiCuCr/C. Compared with the polarization curves obtained in the Ar-saturated electrolyte, clear anodic current density was recorded in the $H_2$-saturated electrolyte, confirming the origin of the anodic current density from HOR (Supplementary Fig. 10). The NiCuCr/C was further tested at different rotating speeds (Fig. 3b), and the anodic current density increased with the increase of the rotating speeds, demonstrating the $H_2$ diffusion-controlled kinetics. Figure 3c shows the fitting of the current density at 50 mV (vs. RHE, the same hereafter) using the Koutechy−Levich equation, and a slope of 4.60 $cm_{disk}^2 \cdot mA^{-1} \cdot s^{-1/2}$ was obtained, which was close to the theoretical value of 4.87 $cm_{disk}^2 \cdot mA^{-1} \cdot s^{-1/2}$ for the $2e^-$ HOR process. The NiCuCr/C exhibited the optimal activity among the four Ni-based materials. Comparable HOR polarization curves in the potential region of 0 - 100 mV could be obtained by using NiCuCr/C with metal loading of 0.079 $mg_M$ $cm^{-2}$ and Pt/C with Pt loading of 0.005 $mg_{Pt}$ $cm^{-2}$ (Fig. 3a), indicating that the synthesized NiCuCr/C was promising to replace Pt/C.

To quantify the HOR activity of the Ni-based catalysts, we further calculated the mass activity. The mass activity was calculated at 50 mV using the kinetic current normalized by total metal loadings (Fig. 3d). The calculation of the kinetic current was described in Supplementary Note 1. The NiCuCr/C showed a high mass activity ($j_{m@50mV}$) of 72.6 A $g_M^{-1}$, which was 2.1, 5.5, and 13.5 times as high as that of NiCu/C, NiCr/C and Ni/C, respectively. This activity is at the highest level in the reported literature (Supplementary Table 2). We also tested NiCuCr/C with different Cu and Cr contents, and it demonstrated that the Cu content of 5% and Cr content of 2% were the optimal (Supplementary Fig. 11, 12), which is the NiCuCr/C composition used in this paper.

To understand the intrinsic activity of the active sites, the electrochemical surface area (ECSA) normalized exchange current density ($j_0$) was calculated. The ECSA of the Ni-based catalysts was estimated

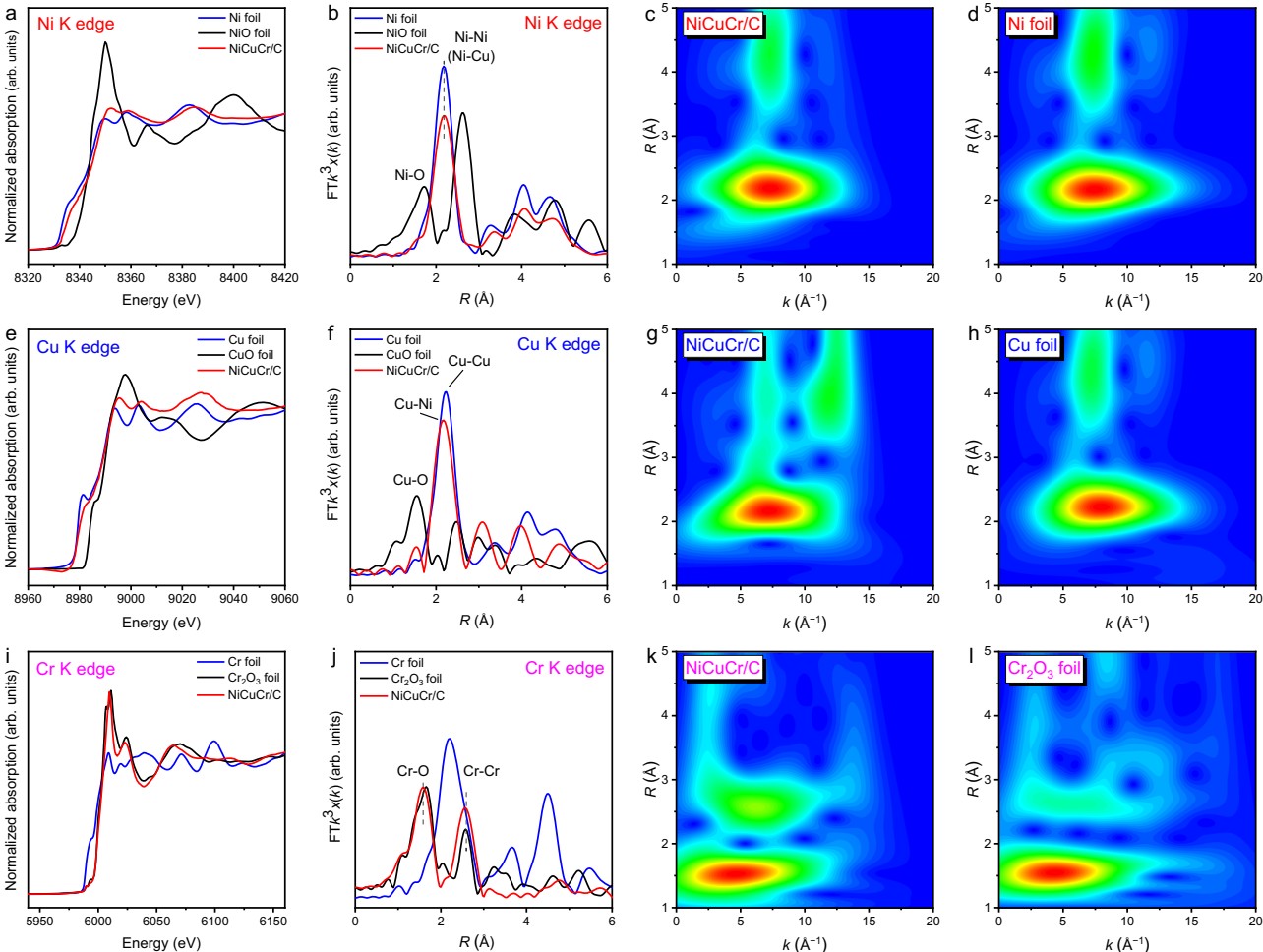

**Fig. 2 | Structural analyses of NiCuCr/C. a** Ni K-edge XANES spectra. **b** Fourier analysis of the EXAFS at the Ni K-edge. **c, d** Corresponding wavelet transforms of $k^2$-weighted EXAFS spectra of Ni K-edge. **e** Cu K-edge XANES spectra. **f** Fourier analysis of the EXAFS at the Cu K-edge. **g, h** Corresponding wavelet transforms of $k^2$-weighted EXAFS spectra of Cu K-edge. **i** Cr K-edge XANES spectra. **j** Fourier analysis of the EXAFS at the Cr K-edge. **k, l** Corresponding wavelet transforms of $k^2$-weighted EXAFS spectra of Cr K-edge. A $k^2$ weighting was applied to amplify the EXAFS oscillations in the mid-$k$ region for all elements.

by the CO-stripping method (Supplementary Fig. 13). The exchange current was calculated from the kinetic current by the Butler-Volmer fitting described in Supplementary Note 1. The NiCuCr/C displayed a high $j_0$ of 135.5 µA $cm_M^{-2}$, which was 6 times as high as that of Ni/C (Fig. 3d) and surpassed most of the reported PGM-free catalysts (Supplementary Table 2).

The stability of NiCuCr/C was evaluated by the accelerated durability test (ADT) using CV cycles between −0.05 and 0.1 V in the $H_2$-saturated electrolyte. After 25,000 CV cycles, the half-wave potential for NiCuCr/C only decreased by 8 mV, indicating good stability (Fig. 3e). Meanwhile, the 5000 CV circled Ni/C showed negligible anodic current, indicating unsatisfactory stability of Ni/C. The NiCuCr/C after the ADT was characterized by TEM and XPS. The TEM image (Supplementary Fig. 14) demonstrated that NiCuCr/C maintained most of its morphology after the ADT test. The elemental mapping images (Supplementary Fig. 15) illustrated Ni, Cu, and Cr distributed in the nanoparticle. The HRTEM image (Supplementary Fig. 16) shows clear lattice fringes of the nanoparticle, and the spacing of the lattice fringe of 0.20 nm was assigned to the (111) facet of Ni, suggesting the NiCuCr/C maintained the metallic Ni form after the ADT test. However, at the surface of the nanoparticle, the spacing of the lattice fringe of 0.24 nm, which was assigned to the (111) facet of NiO, was found, indicating surface oxidation during the ADT. The formation of NiO was confirmed by the SAED (Supplementary Fig. 17) as well, in which an additional diffraction ring assigned to NiO was found. The Ni 2$p$ XPS spectra

(Supplementary Fig. 18) illustrated a larger portion of NiO for NiCuCr/C after the ADT test, confirming the partial oxidation of surface Ni during the ADT. The generated NiO might be the reason for the increased breakdown potential for the NiCuCr/C after the ADT test, as illustrated in Fig. 3e, because oxide substrates had been reported beneficial for Ni-based catalysts against oxidation. However, too much NiO would decrease the HOR activity.

It was known that the operating cost of fuel cells could be reduced by feeding crude hydrogen, which, though low in price[51], usually contains a certain amount of CO impurity[52]. However, CO heavily poisoned the HOR catalysts[53]. We tested the anti-CO poisoning performance of NiCuCr/C and the control catalysts of commercial Ni/C and Pt/C. Figure 3f shows the chronoamperometry at 0.09 V of the catalysts in the electrolyte purging 100 ppm CO-contaminated $H_2$. The current density for NiCuCr/C decreased only 12% after 2500 s of operation, while Ni/C and Pt/C showed a larger current decay (32% and 43%, respectively). The LSV curves of NiCuCr/C before and after 100 ppm CO tests are shown in Supplementary Fig. 19, which indicates that most of the active sites were survived. It indicated the good anti-CO poisoning ability of the NiCuCr/C catalyst.

## High-performance HEMFCs by using the NiCuCr/C as anode catalyst

The synthesized Ni-based catalysts were assembled as anode catalysts in HEMFC MEA. The HEMFC performance was optimized by using

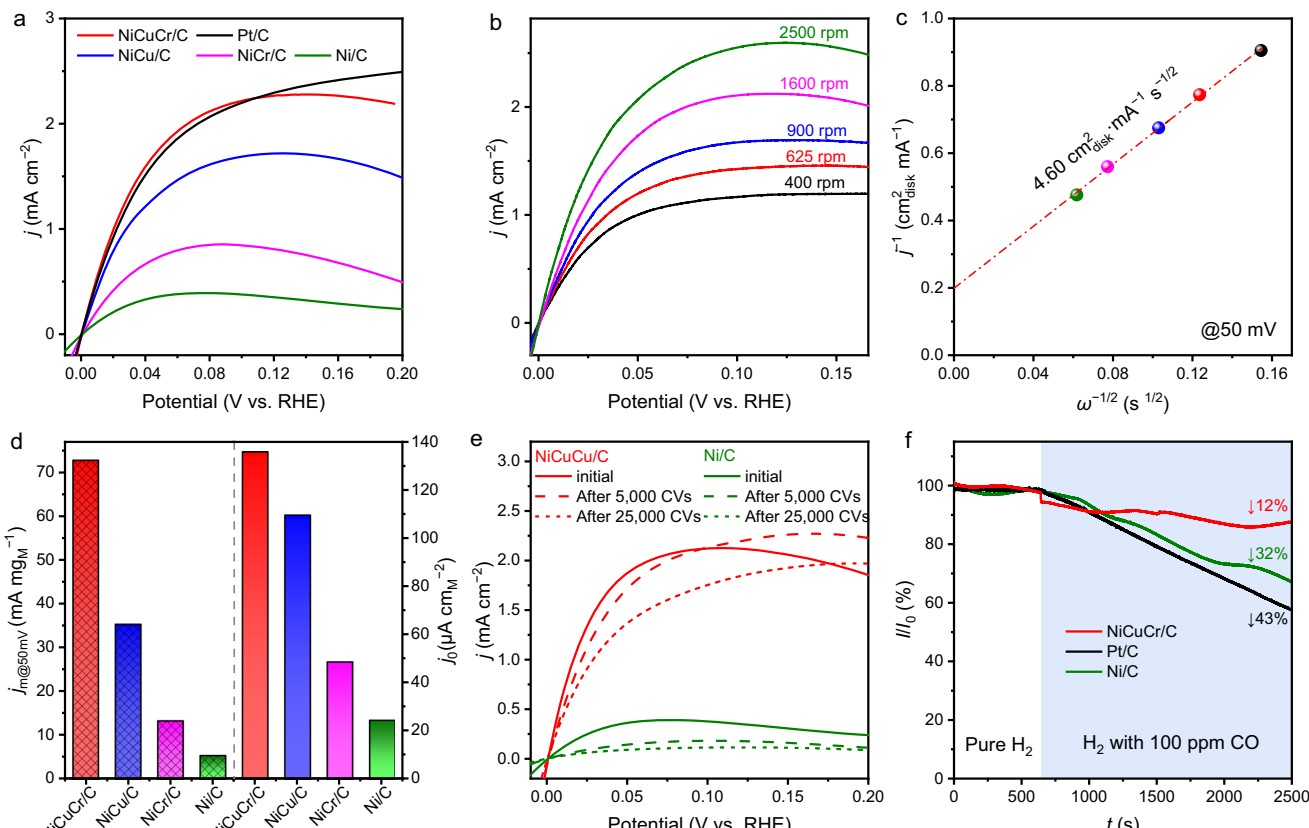

**Fig. 3 | HOR performance tested by RDE. a** Polarization curves of Ni/C, NiCu/C, NiCr/C, NiCuCr/C and commercial Pt/C in $H_2$-satureated 0.1 M KOH at 25 °C. The catalyst loading was 0.078, 0.080, 0.079, and 0.080 $mg_M\ cm^{-2}$ for Ni/C, NiCu/C, NiCr/C, and NiCuCr/C catalyst, respectively. The benchmark Pt/C loading was 0.005 $mg_{Pt}\ cm^{-2}$. The rotating speed was 1600 rpm, and the scan rate was 1 mV s$^{-1}$. **b** Polarization curves of NiCuCr/C in a series of rotating speeds at a scan rate of

1 mV s$^{-1}$. **c** The corresponding Koutecky–Levich plot of NiCuCr/C at 50 mV. **d** The mass activity at 50 mV ($j_{m@50mV}$) and exchange current density ($j_0$) of NiCuCr/C, NiCu/C, NiCr/C and Ni/C. **e** The accelerating durability test result of NiCuCr/C and Ni/C after 5,000 and 25,000 CV cycles. **f** Chronoamperometry test of NiCuCr/C at 0.09 V in pure $H_2$ or $H_2$ containing 100 ppm CO-saturated electrolyte at 25 °C.

different loading of NiCuCr/C (Supplementary Fig. 20), and the optimal anode loading was 9 $mg_M\ cm^{-2}$. Figure 4a shows the polarization curves and power density curves of the HEMFC using a Ni-based anode with $H_2$-$O_2$ feeding. The open circuit of NiCuCr/C HEMFC reached up to 1.06 V, and this cell achieved a high peak power density of 577 mW cm$^{-2}$, which was much higher than those using Ni (32 mW cm$^{-2}$), NiCr (62 mW cm$^{-2}$), and NiCu (102 mW cm$^{-2}$). The cell current density at 0.65 V of NiCuCr/C HEMFC reached up to 880 mA cm$^{-2}$, which was 19.6 times higher than that of Ni/C. The cell performance exceeded most of the Ni-based catalysts that have ever been reported in the literature in recent years (Fig. 4b and Supplementary Table 3). The NiCuCr/C HEMFC even shows competitive cell performance with Pt-based HEMFC at the current density region below 700 mA cm$^{-2}$ (Supplementary Fig. 21). However, at the high current density region, the NiCuCr/C HEMFC still has a significant gap to the Pt-based HEMFCs.

The NiCuCr/C HEMFC demonstrated good stability, represented by the chronoamperometry tests at 0.7 V shown in Fig. 4c. During the 156 h continuous test, the NiCuCr/C HEMFC showed a low average decay rate of 0.6 mA cm$^{-2}$ h$^{-1}$. By comparison, the Ni/C HEMFC decayed at a higher rate of 4.1 mA cm$^{-2}$ h$^{-1}$ at initial, and the current density suddenly descended to tens of mA cm$^{-2}$ within 20 h of tests. The NiCuCr/C HEMFC is the most stable anode PGM-free HEMFC reported in the literature (Fig. 4b). NiCuCr nanoparticles showed no obvious agglomeration and phase transition after the 156 h of operation, as demonstrated by the TEM image and XRD pattern (Supplementary Fig. 22). The Ni 2$p$ XPS spectra (Supplementary Fig. 23a) illustrated an

increased oxidized Ni portion after the stability test, and the Cr 2$p$ XPS spectra (Supplementary Fig. 23b) demonstrated that the Cr maintained as Cr$^{3+}$. The Ni/Cu/Cr ratio measured from XPS was 1/0.165/0.081 at the initial and 1/0.089/0.080 after the stability test. It demonstrated that most of the Cr survived in the stability test, but partial Cu was leached. The possible Pt migration from cathode to anode during the long-term test was excluded by the XPS results of the anode after the durability test, which did not show any Pt signals (Supplementary Fig. 24).

Encouraged by the high MEA performance of the NiCuCr/C anode, we fabricated a totally PGM-free MEA by using the NiCuCr/C as the anode and commercial Ag/C as the cathode. Ag showed high ORR performance in alkaline conditions but much lower cost than PGMs[54–57]. High-performance HEMFCs using Ag as the cathode have been reported[21,58,59]. Figure 4d shows the polarization curves and power density curves of the totally PGM-free MEA with $H_2$-$O_2$ feeding under different back pressures. It reached peak power densities of 335 and 264 mW cm$^{-2}$ at 2.0 and 1.0 bar back pressure, respectively. The peak power density of NiCuCr-Ag totally PGM-free HEMFC in this work surpassed most of the totally PGM-free fuel cells reported in the literature (Fig. 4e and Supplementary Table 4).

### Tuning the apparent binding energy to achieve high-performance Ni-based HOR catalysts

The experiments above demonstrated the superb HOR activity and stability of the NiCuCr/C catalyst in both the RDE test and MEA. The high-performance was attributed from its unique electronic and

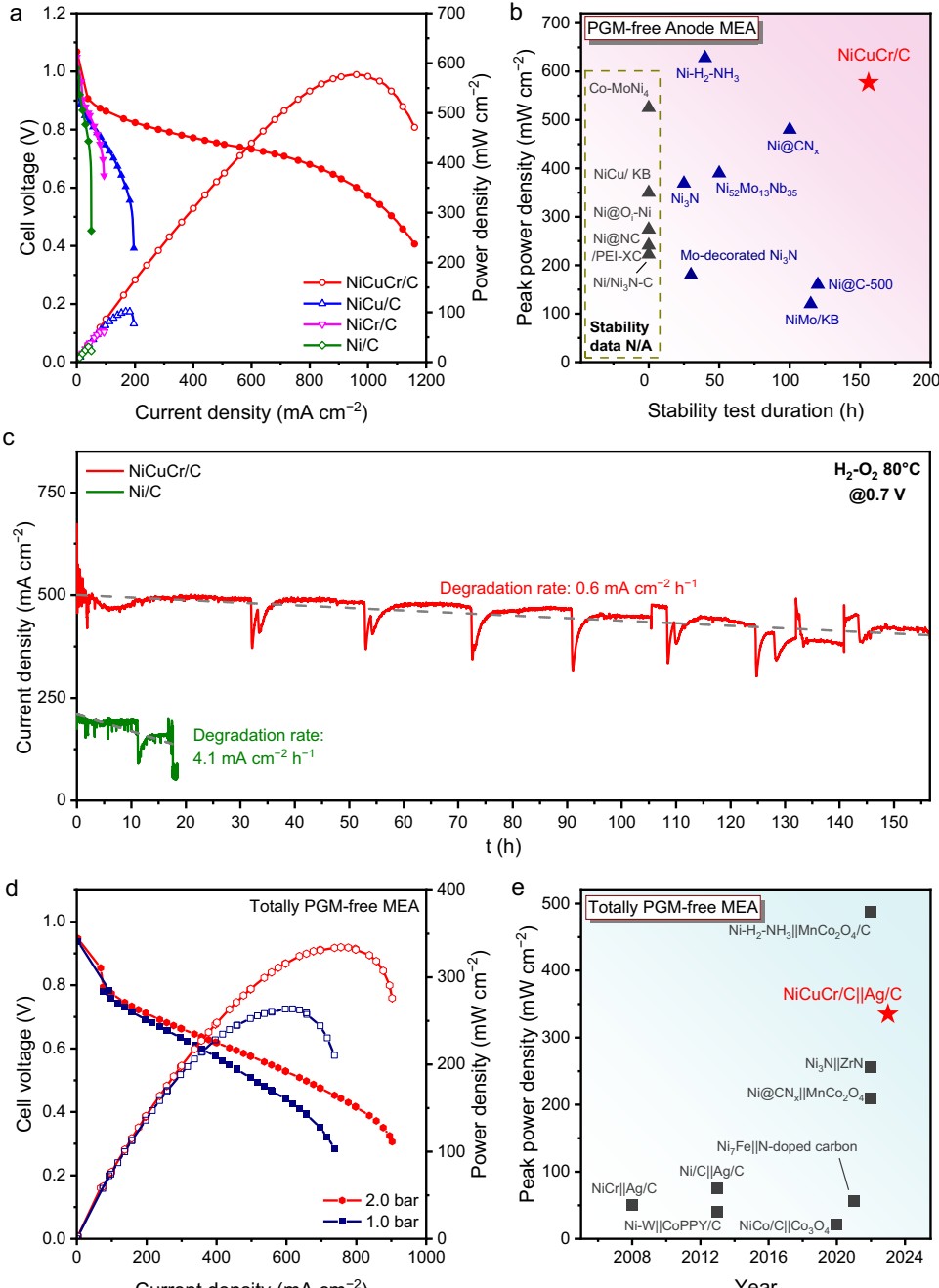

**Fig. 4 | HEMFC performance and durability. a** $H_2$-$O_2$ HEMFC polarization and power density curves with Ni/C, NiCu/C, NiCr/C, or NiCuCr/C anode. The anode catalyst loading was 9 $mg_M$ $cm^{-2}$, and the cathode catalyst loading was 0.2 $mg_{Pt}$ $cm^{-2}$ using commercial 40% Pt/C. The anode humidifier temperature, cathode humidifier temperature, and cell temperature were 78, 80, and 80 °C, respectively. The back pressure was 2.0 bar, and the flow rates of $H_2$ and $O_2$ were both 1000 sccm. **b** Summary of the peak power density and durability of the reported PGM-free anode MEA, details are listed in Supplementary Table 3. **c** $H_2$-$O_2$

HEMFC stability test curves of NiCuCr/C and Ni/C at 0.7 V, the test condition was the same as polarization test except for the flow rate of $H_2$ and $O_2$ were both 500 sccm. **d** $H_2$-$O_2$ HEMFC polarization and power density curves of the MEA using NiCuCr/C anode and Ag/C cathode. The anode catalyst loading was 9 $mg_M$ $cm^{-2}$, and the cathode catalyst loading was 0.6 $mg_{Ag}$ $cm^{-2}$ using commercial 20% Ag/C. **e** Summary of the peak power density of reported totally PGM-free MEA; details are listed in Supplementary Table 4.

surface structures. The HOR process underwent the adsorption of the $H_2$ on the surface of the catalysts to form *H. Then, the *H transferred electron was released into the electrolyte, and the surface was recovered by water. Thus, the apparent binding energies to the intermediate *H, i.e., $HBE_{app}$, controlled the HOR process. The "volcano" plot indicated that Ni had a stronger intrinsic HBE than the optimum, and thus optimizing its $HBE_{app}$ required the weakening of the HBE and strengthening of the WBE.

The HBE of the catalysts was controlled by their band structures, and it could be adjusted by the alloying effect. The band structures of the Ni-based catalysts were studied by ultraviolet photoelectron spectroscopy (UPS). Figure 5a shows the filled valence band spectra of Ni, NiCu, NiCr, and NiCuCr, and the bands were observed adjacent to the Fermi level for all the samples, which were ascribed to the Ni 3$d$ states represented to the $d$-band structure[60]. By using the Fermi level (0 eV) as a reference point, their peak positions were 0.15, 0.40, 0.20,

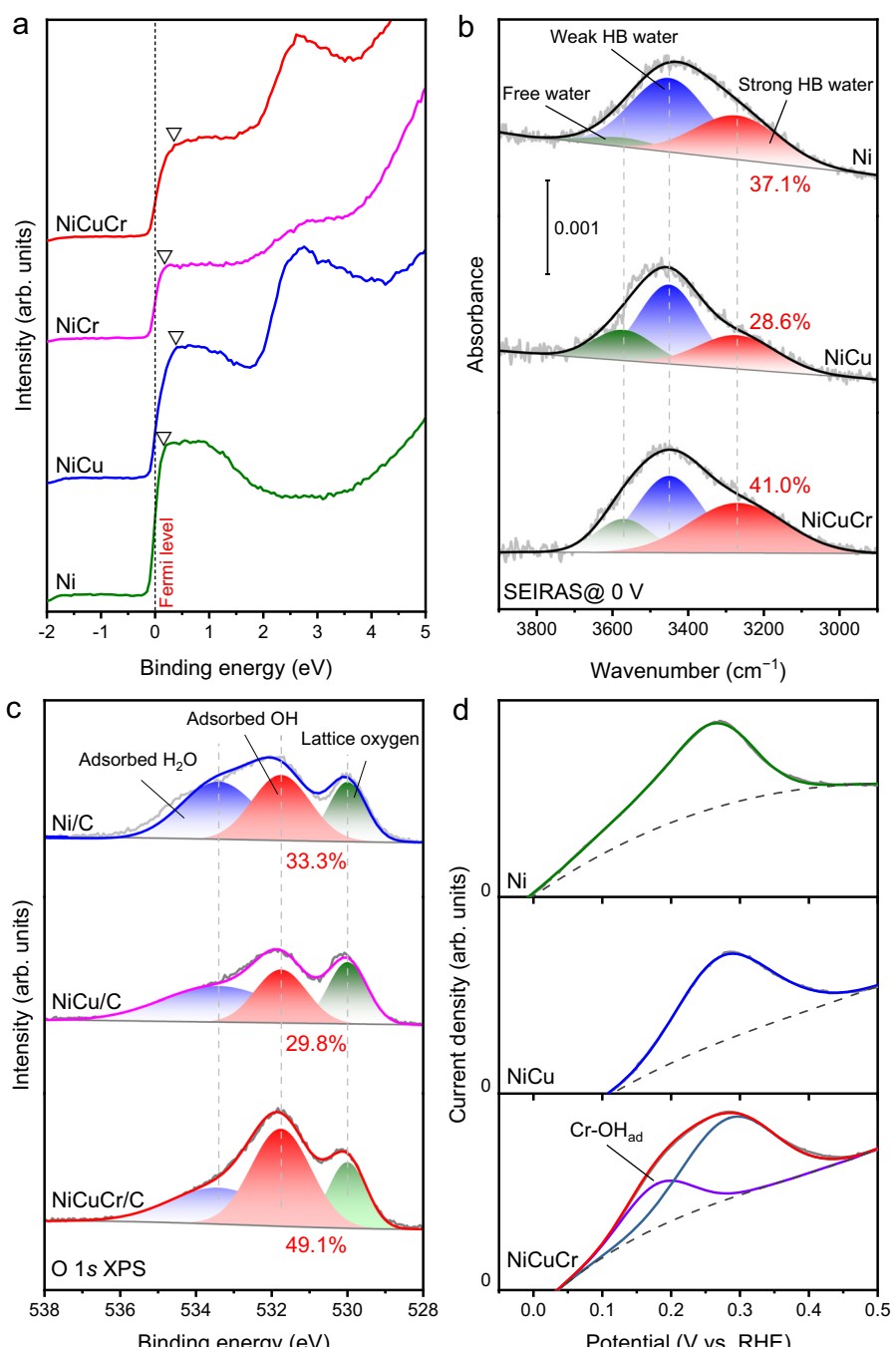

**Fig. 5 | Catalyst surface characterizations. a** UPS spectra of Ni, NiCu, NiCr, and NiCuCr. **b** Electrochemical in situ ATR-SEIRAS of Ni, NiCu, and NiCuCr. The spectra were obtained at 0 V. **c** O 1$s$ XPS spectra of Ni/C, NiCu/C, and NiCuCr/C. **d** Anodic LSV in Ar-saturated 0.1 M KOH at a scan rate of 50 mV s⁻¹.

and 0.37 eV for Ni, NiCu, NiCr, and NiCuCr, respectively. It illustrated that the addition of Cu can efficiently lower the $d$-band to the Fermi level. According to the $d$-band theory, the HBE was correlated to the filling statues of the metal-H antibonding states[61], and a lower metal $d$-band was regarded as a weakened HBE, which was beneficial for promoting the HOR activity of Ni-based catalysts. This result demonstrated that alloying with Cu could efficiently weaken the HBE, thus improving the HOR activity. However, the NiCu and NiCuCr illustrated similar $d$-band levels, but they displayed many different HOR activities, implying additional factors associated with HOR activity.

From the view of the HBE$_{app}$ theory, the HBE$_{app}$ accounts for not only the inherent HBE but also the binding of the interfacial water[47]. We investigated the interfacial water structures through electrochemical

in situ attenuated total reflection surface-enhanced infrared reflection absorption spectroscopy (ATR-SEIRAS) on Ni, NiCu, and NiCuCr in the potential windows −0.5 ~ 0.2 V in H$_2$-saturated 0.1 M KOH. As is shown in Supplementary Fig. 25, the OH stretching (~3600 ~ 3200 cm⁻¹) from the interfacial water was successfully probed. Clearly, with the additional elements, the profile of the OH stretching peak alternated, demonstrating the different interfacial water binding strength. The OH stretching peaks can be deconvoluted into three peaks, which were free water (3570 cm⁻¹), weakly H-bonded water (3450 cm⁻¹), and strongly H-bonded water (3270 cm⁻¹), with the different bonding strengths[46,62]. The proportions of different types of water on different catalysts' surfaces at 0 V were compared in Fig. 5b. Ni had 37.1% of strongly H-bonded water. After the introduction of Cu, the strongly

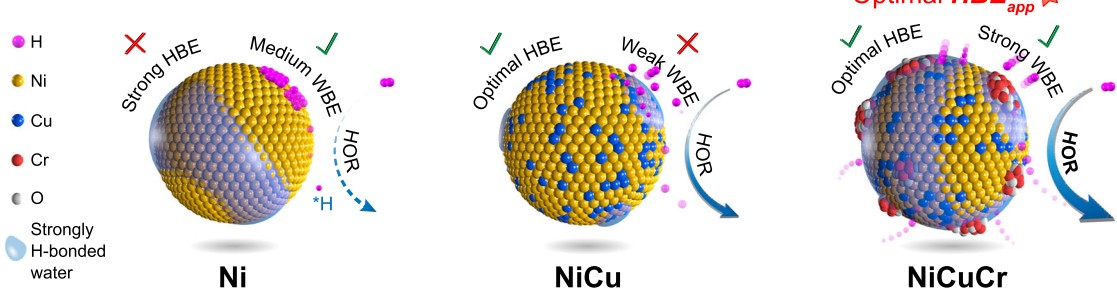

**Fig. 6 | Scheme illustration of catalyst characteristics.** The hydrogen binding energy (HBE), water binding energy (WBE), and finally apparent hydrogen binding energy (HBE$_{app}$) features of Ni/C, NiCu/C, and NiCuCr/C.

H-bonded water ratio decreased to 28.6%, resulting in the overall water peak positively shifting by 25 cm$^{-1}$. It demonstrated that although alloying Ni with Cu optimized its HBE, its WBE deteriorated. When Cr was introduced as the NiCuCr, a clearly increased strongly H-bonded water ratio up to 41.0% was observed, higher than that for pristine Ni and NiCu. It demonstrated that the cooperation of Cr could remarkably strengthen the water binding. This strengthened effect was exhibited in the whole tested potential range (−0.5 - 0.1 V, Supplementary Fig. 26, 27).

The strengthened WBE of NiCuCr was attributed to the introduction of the hydrophilic Cr. Based on the catalyst characterization results, the Cr enriched on the surface of the NiCuCr nanoparticles and existed as Cr$_2$O$_3$. Compared with Ni and Cu, Cr was easier to adsorb OH$^-$[63,64]. The more OH on the surface was proved by the O 1s XPS spectra (Fig. 5c). The peaks at 533.4, 531.8, and 530 eV were assigned to the adsorbed water, adsorbed OH, and lattice oxygen, respectively. The lattice oxygen originated from the surface oxides because the samples were stored in the air. A higher portion of adsorbed OH was observed for the NiCuCr catalysts, demonstrating the rich surface OH beneficial from the hydrophilic Cr. The surface OH can connect the water molecule well and thus improve the WBE of the catalysts. Chen et al.[45] has demonstrated that the surface OH was helpful to interact with the interfacial water and thus constructed a better H-bond network, which was critical for the HOR process. The adsorbates on the catalyst surface were investigated by CVs as well. As shown in Fig. 5d, the anodic curves represent the OH adsorption. Compared with Ni, alloying Cu deterred the Ni(OH)$_2$ peak potential by 23 mV, which may be caused by the changed band structure of the NiCu, as discussed earlier. The lower d-band for NiCu weakened not only the HBE but also OHBE. However, NiCuCr showed an extra peak at a lower potential of about 0.18 V, demonstrating the stronger interaction achieved by the heterostructured surface Cr$_2$O$_3$ clusters. This peak originated in OH adsorption on Cr$_2$O$_3$ species, as proved by the CV curve of Cr$_2$O$_3$/C (Supplementary Fig. 28).

## Discussion

Based on the above results, the NiCuCr/C catalyst has weakened HBE through the alloying effect of additional Cu and strengthened WBE through the heterostructured surface Cr$_2$O$_3$ clusters. Both effects contributed to the weakened HBE$_{app}$ to an optimized condition for HOR. The weakened HBE resulted in the smoother electron transfer of the *H, and the strengthened water binding promoted the proton transfer and made it easier to combine with OH$^-$ to generate water. These effects promoted the kinetics of the Volmer step, which was considered the rate-determining step for the HOR on Ni. A schematic illustration of the optimizing effect of the HBE$_{app}$ for NiCuCr/C is shown in Fig. 6.

In conclusion, a two-pronged strategy was successfully applied to construct a highly active NiCuCr/C HOR catalyst. The Ni nanoparticle was optimized by component regulation of alloying with Cu, which

brings the catalyst to have weakened HBE and higher oxidation resistance. By decorating Cr oxides on the surface of the NiCu nanoparticles, strengthened WBE was achieved. The contrary regulation direction was achieved by the heterostructured nanoparticles, and both effects tuned toward the optimized HBE$_{app}$. The as-obtained NiCuCr/C catalyst showed a high-performance in the H$_2$-O$_2$ HEMFC test, reaching 577 mW cm$^{-2}$ in peak power density and surviving in 156 h of durability test. These results showed that fine-tuning the HBE$_{app}$ was an effective approach to promote the HOR catalysts, and the synthesized NiCuCr/C was promising as an anode catalyst for HEMFCs.

## Methods

### Synthesis of NiCuCr/C catalysts

The NiCuCr/C was synthesized by impregnation. 300 mg of Ketjen black was dispersed in 15 mL of ethanol in a round bottom flask, and the slurry was magnetically stirred for 30 min. 3.57 g of Ni(NO$_3$)$_2$·6H$_2$O, 0.148 g of Cu(NO$_3$)$_2$·3H$_2$O, 0.082 g of Cr(NO$_3$)$_2$·9H$_2$O, 0.023 g of urea and 0.207 g of citric acid monohydrate were dissolved in 3 mL of deionized water. Then, the solution was added to the carbon slurry dropwise. The obtained slurry was magnetically stirred for 10 h at room temperature and then dried at 80 °C in an oven. Finally, it was placed in a furnace and calcinated at 470 °C for 1 h under the atmosphere of 5% H$_2$ and 95% Ar with a gas flow rate of 50 sccm.

The control samples of Ni/C, NiCu/C, and NiCr/C were synthesized following a similar procedure but only adding the corresponding metal salts.

### Physical characterizations

Transmission electron microscopy (JEOL JEM1230) and high-resolution transmission electron microscopy (JEOL JEM1230, equipped with an energy dispersive X-ray spectrometer) were used to determine the morphologies of the samples. The measurements of the X-ray absorption spectra (XAS) were performed at the TPS-21A beamline of the national synchrotron radiation research center (NSRRC, Hsinchu, China) using a Si (111) quick-scanning monochromator. Data processing of the X-ray absorption near edge structure (XANES) and the extended X-ray absorption fine structure (EXAFS) were carried out using Athena and Artemis modules (Version 0.9.26). The XPS was acquired by Thermo Fisher ESCALAB 250Xi spectrometer equipped with a monochromatic Al K$_\alpha$ X-ray source. The X-ray diffraction (XRD) data were obtained on a Rigaku D/Max 2500 VB2+/PC X-ray powder diffractometer equipped with a monochromatic Cu K$_\alpha$ source with a scan rate of 10° min$^{-1}$. Inductively coupled plasma-optical emission spectroscopy (ICP-OES, Thermo Fisher iCAP PRO) was used to determine the compositions of the samples.

### Electrochemical measurements

Electrochemical measurements were performed in a standard three-electrode cell system using an electrochemical workstation (V3,

Princeton Applied Research) combined with a rotating disk electrode system (RDE, PINE Instruments) in 0.1 M KOH electrolyte. The polished RDE (5 mm in diameter) were served as the substrate for supporting catalysts. A saturated calomel electrode (SCE) and a graphite rod were employed as the reference electrode and counter electrode, respectively. The catalyst ink was prepared by dispersing 5 mg of catalyst in a mixed solution consisting 100 μL of water, 895 μL of isopropanol, and 5 μL of Nafion suspension (Du Pont, 5 wt %). The above suspension was sonicated for 2 h in an ice bath to form catalyst ink. Then, 5 μL of the catalyst ink was dropped onto the polished glassy carbon electrode with a total catalyst loading (including metal and carbon support) of 0.128 mg cm$^{-2}$. Commercial Pt/C (Alfa Aesar, 20 wt %) was used as a reference and the Pt loading was 0.005 mg$_{Pt}$ cm$^{-2}$. All the potentials were reported to the reversible hydrogen electrode (RHE) scale. To calibrate the zero point of RHE, 20% Pt/C was used as working electrode to measure equilibrium potential of HER/HOR in a H$_2$-saturated electrolyte. Linear sweep polarization curves were scanned at 1 mV s$^{-1}$ and compensated using $iR$ correction. Electrochemical impedance (100 kHz–1 Hz) measurement was conducted to obtain the resistance of electrolyte ($R$). All the electrochemical measurements were conducted at 25 °C.

The CO tolerance measurement were conducted using chronoamperometry at 0.09 V in the presence of 100 ppm CO. For the CO-stripping test, CO gas was firstly bubbled into the electrolyte and the working electrode was held at 0.1 V for 10 min. Then the electrolyte was flushed with Ar for 20 min to remove the CO in electrolyte. Subsequently, the adsorbed CO was stripped by scanning between 0.05 and 1.0 V at a scan rate of 50 mV·s$^{-1}$.

## Electrochemical in situ ATR-SEIRAS test

The in situ ATR-SEIRAS experiments were taken with Nicolet iS50 FT-IR spectrometer equipped with an MCT detector. The spectral resolution was set as 4 cm$^{-1}$ and 64 interferograms were co-added for each spectrum. A PIKE VeeMAX III variable angle ATR sampling accessory was employed to done the ATR tests. A Si face-angled crystal with the incident angle of 60° was used as the reflection element. Au film was deposited chemically on it for IR enhancement and electronic conduction. The Ni, NiCu and NiCuCr was electrochemically deposited onto the Au film at −2.0 V (vs. Ag/AgCl) to serve as the working electrode for SEIRAS experiments. SEIRAS spectra were collected at potential from −0.5 to 0.2 V. The reference single beam spectrum was collected at 0.2 V. The spectra were given in absorbance defined as Abs = −lg($R/R_0$), where $R$ and $R_0$ represent the reflected IR intensities corresponding to the sample and reference single beam spectrum, respectively.

## MEA tests

Fuel cell tests were conducted by measuring the polarization curve of MEA with an active area of 5 cm$^2$. The MEA was prepared using a catalyst-coating membrane method. Typically, NiCuCr/C was employed as the anode catalyst, and commercial Pt/C was used as the cathode catalyst. The catalyst ink was prepared by ultrasonically dispersing the catalyst and ionomer (PAP-TP-100, 3.5 wt% in ethanol, Versogen) into water and isopropanol (1:20 v/v) for 1 h. The anode ink was sprayed onto one side of the membrane (PAP-TP-85, Versogen) with a loading of 9 mg$_M$ cm$^{-2}$ and the cathode ink was sprayed onto the other side of the membrane with a loading of 0.4 mg$_{Pt}$ cm$^{-2}$ for Pt/C (or 0.6 mg$_{Ag}$ cm$^{-2}$ for Ag/C). Three airbrushes were used for Ni-based, Pt-based, and Ag-based catalysts separately to prevent possible cross-contamination. The gas source of the anode airbrush was switched to N$_2$ to prevent possible oxidative damage to the Ni-based materials. The GDLs for both anode and cathode were SGL-29 BC from SGL. The MEA was immersed in 2 M KOH for 3 h and then rinsed thoroughly with deionized water before tests. The fuel cell performance was tested on a fuel cell test station (850e, Scribner) equipped with a back pressure module.

## Data availability

The data generated in this study are provided in the Source Data file. Source data are provided with this paper.

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

## Acknowledgements

This work was financially supported by the National Key Research and Development Program of China (2019YFA0210300, Z.Z.), the National Natural Science Foundation of China (22379004, Z.Z.), Beijing Natural Science Foundation (Z210016, Z.Z.), and Fundamental Research Funds for the Central Universities (buctrc201916, Z.Z.).

## Author contributions

Z.Z. and X.W. conceived the ideas. X.W. synthesized the catalysts. Ho.W. collected and analyzed the XAS data. X.W., Xi.L., Ha.W., and X.Z. performed catalyst characterizations and the RDE tests. X.W., Xu.L., and Y.W. performed the MEA tests. J.F. and C.C. carried out the in situ ATR-SEIRAS experiments. W.Z. helped with data interpretation. X.W., J.F., and Z.Z. wrote the manuscript with support from all co-authors.

## Competing interests

The authors declare no competing interests.
