## [Peer Review File · Nature Communications]

REVIEWER COMMENTS

Reviewer #1 (Remarks to the Author):

In this work, NiCuCr catalysts composed of NiCu alloy particles with Cr₂O₃ clusters dispersed on the surface were prepared for alkaline hydrogen oxidation. The alloying effect from NiCu weakens the hydrogen binding and the surface Cr₂O₃ species enhance the interfacial water binding, which results in optimized apparent hydrogen binding energy and thus leads to the high HOR performance. In my view, this work requires major revision before it can be considered as a publication.

1. The authors describe electron transfers from Ni to Cu or Cr additives in NiCuCr/C catalysts. Apart from the change of Ni XPS spectra, please provide the Cu or Cr XPS spectra for NiCu/C or NiCr/C to prove this conclusion.
2. NiCuCr/C catalysts show higher breakdown potential after the stability test in Figure 3e, why? Meanwhile, it is suggested to provide the structure and composition results of NiCuCr/C after stability.
3. The lower onset potential of CO stripping is used to prove the strengthened OH adsorption on NiCuCr/C in Figure S18. However, the CO oxidation peak of NiCuCr/C is shift to a higher potential compared with that of Ni/C, indicating a weakened OH adsorption. Please give a reasonable explanation.
4. It is suggested to explore the effect of Cu contents on HOR performance.
5. The Cr₂O₃ species enhance the interfacial water binding, How about the amount of Cr₂O₃ on the HOR performance?
6. NiCuCr/C shows high performance in HEMFC, how does it compare with commercial Pt/C?

Reviewer #2 (Remarks to the Author):

The key finding in this work is the unique roles that Cu and Cr plays in the NiCuCr nano-electrocatalyst for HOR. The Cu weakens the hydrogen binding while the Cr strengthens the interfacial water binding, both of which enhance the performance of HOR. The findings were used to substantiate the reasons why the catalyst performed better than many previous literature on the same subject of HOR with PGM-free catalysts.

The work is well conducted, well written, with all interpretations based on the experimental and theoretical findings and should be considered for publication in the Nature Communications after minor revisions.

1. While we understand the reason for Ni, what informed the choice of Cu and Cr in this work? Would other metallic elements in the same periodic table exhibit same properties?

2. Figure 5C, what is the source of the lattice oxygen? It seems everything is attributed to water, why? Could the alloys have some oxide phases? It may not be possible to remove all the oxide by 1 h annealing under 5% H₂ / 95% Ar.

3. What about the XPS of the carbon support? Also, why is it not seen in the XRD?

4. Check the grammar. For example, p.4 line 55 "significantly influent" is a wrong grammar. please correct.

Reviewer #3 (Remarks to the Author):

In the manuscript by Wang et al., the authors report the NiCuCr/C electrocatalyst for the hydrogen oxidation reaction in alkaline conditions. A high activity was observed for the NiCuCr/C in RDE test, and the MEA using the NiCuCr/C demonstrated high peak power density as well, illustrating the high promising of this reported catalyst. The authors attributed the high performance to the optimized apparent hydrogen binding, which coming from the alloy effects and the surface oxides. Some in-situ characterizations, such as ATR-SERIES was employed to investigate the reaction mechanism.

The presented data are well organized and the manuscript was well written, I would recommend the manuscript for publication in Nature Communications after solving the following issues.

1. The Cr₂O₃ cluster on the surface was considered important for the tuning of the apparent hydrogen binding energy. However, the Cr₂O₃ was possible to be dissolved in alkaline condition. I wonder are the Cr₂O₃ survived after HEMFC test? What is the chemical state of Cr after stability test? Please specify in the manuscript.

2. In HEMFC stability test (Figure 4c), sudden drops of cell voltage were observed. What was the reason? Did they come from the degradation of the catalysts?
3. PtRu/C was generally used as the benchmark anode catalyst for the anion exchange membrane fuel cells. It would be better if the fuel cell performance of the catalyst is compared with the PtRu/C, although the Ni-based catalysts always have worse performance than the PGM catalysts.
4. The study regarding stability for Ni based catalyst (ACS Appl Mat Interfaces 12, 31575-31581 (2020)) should be cited and added the corresponding data in Figure 4b.
5. The Ag/C was used as the cathode to assemble non-Pt MEA. Why Ag/C was selected as one of the precious metals? Please add more explanation in the manuscript.
6. Some experimental details are missing, for example, the test temperature for the LSV and chronoamperometry curve shown in Figure 3, the MEA test condition for the stability shown in Figure 3c.

RESPONSE TO THE REVIEWERS' COMMENTS

We sincerely thank all the reviewers for careful attention to our manuscript and valuable, constructive comments. The comments from reviewers are copied in *italics* and our point-by-point responses to each comment are given in blue.

Reviewer #1:

General Comments: *In this work, NiCuCr catalysts composed of NiCu alloy particles with Cr₂O₃ clusters dispersed on the surface were prepared for alkaline hydrogen oxidation. The alloying effect from NiCu weakens the hydrogen binding and the surface Cr₂O₃ species enhance the interfacial water binding, which results in optimized apparent hydrogen binding energy and thus leads to the high HOR performance. In my view, this work requires major revision before it can be considered as a publication.*

Response: Thanks for your great efforts in reviewing our manuscript. All of your comments are highly important and valuable for us to improve the manuscript. We addressed the comments point-by-point and made the corresponding changes accordingly in the revised manuscript.

Comment 1. *The authors describe electron transfers from Ni to Cu or Cr additives in NiCuCr/C catalysts. Apart from the change of Ni XPS spectra, please provide the Cu or Cr XPS spectra for NiCu/C or NiCr/C to prove this conclusion.*

Response: Thanks for your valuable suggestion and we apologize for the unclear description of the XPS results. We tested the Cu or Cr XPS spectra for NiCu/C or NiCr/C, and the results are shown in Figure R1. Compared with NiCu/C, NiCuCr/C present a blue shift of 0.3 eV in the Cu 2p spectra, indicating the tuning effect of the Cr additives. For the Cr 2p spectra, NiCr/C and NiCuCr/C showed similar peak positions, which were assigned to Cr(III) form. We have revised the discussion of the XPS results as follows.

Figure R1. (a) High resolution Cu 2p XPS spectra of NiCuCr/C and NiCu/C. (b) High resolution Cr 2p XPS spectra of NiCuCr/C and NiCr/C.

Revision: Page 7, line 126. “The Ni (0) peaks of NiCuCr/C, NiCu/C and NiCr/C show a positive shift (~0.8 eV) compare with Ni/C, indicating the transferred electrons from Ni. Supplementary Figure 6 shows the high-resolution Cu 2*p* spectra of NiCuCr/C and NiCu/C, indicating the Cu mainly existed in Cu (0) state. The NiCuCr/C illustrates a negative shift of ~0.3 eV compared with the NiCu/C, demonstrating the tuning effect of Cu additives. Supplementary Figure 7 shows the high-resolution Cr 2*p* spectra of NiCuCr/C and NiCr/C. They displayed two similar peak positions at 577.5 and 587.3 eV, indicating that the Cr was mostly in oxidized Cr(III) form.”

Comment 2. *NiCuCr/C catalysts show higher breakdown potential after the stability test in Figure 3e, why? Meanwhile, it is suggested to provide the structure and composition results of NiCuCr/C after stability.*

Response: We thank the reviewer for the valuable question and constructive suggestion. Ni based HOR catalysts are poisoned at high potential, which make the polarization curves break down when scanning to high potential. As demonstrated in Figure 3e, the breakdown potential of the NiCuCr/C increased after stability tests. The possible reason is NiO_x were generated during the stability tests. It had been reported that cooperating with oxide, the *d* band center of Ni could be adjusted, leading to a weakened adsorption of the oxygen intermediate. Thus, the adequate oxide coverage was beneficial for increase of the breakdown potential of the HOR catalysts (*ACS Appl Energy Mater* **2**, 5534-5539 (2019); *J Electrochem Soc* **167**, 084517 (2020); *Nat Catal* **3**, 454-462 (2020)).

We characterized the NiCuCr/C after stability test (denoted as NiCuCr/C-After ADT) by HRTEM and XPS. As shown in Figure R2a, the TEM image of NiCuCr/C-After ADT shows NiCuCr nanoparticles distributed uniformly on the carbon support, similar to the initial NiCuCr/C. The corresponding HRTEM image (Figure R2b) shows the spacing of the lattice fringe of not only 0.20 nm assigned to Ni (111) facets, but also 0.24 nm assigned to NiO (111) facets, revealing the formation of NiO clusters of the edge of NiCuCr. Compared with the initial NiCuCr/C, the NiCuCr/C-After ADT illustrated an additional weak diffraction ring in the selected area electron diffraction pattern (SAED, Figure R2c), which was assigned to NiO (220). It suggested that the NiO were generated during the durability tests. The elemental mapping images illustrated Ni and Cu are distributed homogeneously in the nanoparticle while the Cr and O are concentrated on the external surface of nanoparticle (Figure R2e).

Figure R2. (a) TEM image of NiCuCr/C-After ADT. The insert is the statistical histogram of the size of the particles. (b) HRTEM image of NiCuCr/C-After ADT. Selected area electron diffraction pattern of (c) NiCuCr/C and (d) NiCuCr/C-After ADT. (e) HAADF-STEM elemental mapping image of NiCuCr/C-After ADT.

We also tested the XPS of the NiCuCr/C-After ADT. As shown in Figure R3, the Ni 2p spectra were deconvoluted into four subpeaks. The peak at 853.1, 854.3, 856.1 and 861.8 eV were assigned to Ni⁰, NiO, Ni(OH)₂ and Ni satellite, respectively. Compared with NiCuCr/C, the NiCuCr/C-After ADT clearly displayed larger portion of NiO on the surface, confirming the partial oxidation of surface Ni during the ADT.

Figure R3. High resolution Ni 2p XPS spectra of NiCuCr/C-After ADT and NiCuCr/C.

Revision: Page 12, line 220. “The NiCuCr/C after the ADT was characterized by TEM and XPS. The TEM image (Supplementary Figure 13) demonstrated NiCuCr/C maintained most of its morphology after the ADT test. The elemental mapping images (Supplementary Figure 14) illustrated Ni, Cu and Cr distributed in the nanoparticle. The HRTEM image (Supplementary Figure 15) shows clear lattice fringes of the nanoparticle, and the spacing of the lattice fringe of 0.20 nm was assigned to (111) facet of Ni, suggesting the NiCuCr/C maintained the metallic Ni form after ADT test. However, at the surface of the nanoparticle, the spacing of the lattice fringe of 0.24 nm was found, which was assigned to (111) facet of NiO, indicating the surface oxidation during the ADT. The formation of NiO was confirmed by the SAED (Supplementary Figure 16) as well, in which an additional diffraction ring assigned to NiO was found. The Ni 2p XPS spectra (Supplementary Figure 17) illustrated larger portion of NiO for NiCuCr/C after the ADT test, confirming the partial oxidation of surface Ni during the ADT. The generated NiO might be the reason for the increased breakdown potential for the NiCuCr/C after ADT test, as illustrated in Figure 3e, because oxide substrates have been reported beneficial for Ni-based catalysts against oxidation. However, too many NiO will decrease HOR activity.”

Comment 3. *The lower onset potential of CO stripping is used to prove the strengthened OH adsorption on NiCuCr/C in Figure S18. However, the CO oxidation peak of NiCuCr/C is shift to a higher potential compared with that of Ni/C, indicating a weakened OH adsorption. Please give a reasonable explanation.*

Response: We thank the reviewer for pointing it out. We apologize for the unclear description in the original manuscript. The samples shown in Figure S18 were the electrochemically deposited thin films. The sites on NiCuCr were complex because of its multi-component nanoparticle structure. The CO stripping curves displayed wide CO oxidation peaks, demonstrating different sites on NiCuCr as well. The NiCuCr illustrated lower onset potential of CO stripping, indicating the NiCuCr has some strong OH adsorption sites. However, the NiCuCr also displayed higher peak position, indicating the NiCuCr also

has some weak OH adsorption sites. For a catalytic process, the overall catalytic activity was always mainly contributed by the sites with high activity (*Nat Rev Chem* **6**, 89-111 (2022)). The NiCuCr has some strong OH adsorption sites, which has strong interaction with water, and thus facilitating the HOR process.

To avoid misunderstanding, we decided to remove the Figure S18 and the related discussion in the revised manuscript.

Comment 4. *It is suggested to explore the effect of Cu contents on HOR performance.*

Response: We thank the reviewer's constructive suggestion. The effect of Cu content to the HOR activity of NiCuCr/C was studied. The Cu content was adjusted by changing the precursor ratio between $\text{Cu}(\text{NO}_3)_2 \cdot 3\text{H}_2\text{O}$ and $\text{Ni}(\text{NO}_3)_2 \cdot 6\text{H}_2\text{O}$. As shown in Figure R5a, the HOR activity varies according to different Cu/Ni molar ratios. To quantify the HOR activity of the Ni-based catalysts, we calculated the exchange current density and mass activity at 50 mV, and the results were summarized in Figure R5b. The optimal Cu/Ni ratio was found to be 0.05.

Figure R5. (a) Polarization curves of NiCuCr/C catalyst with different Cu/Ni ratios in H₂-saturated 0.1 M KOH. The catalyst loading was 0.080 mg_M cm⁻² for catalysts. The rotating speed was 1600 rpm and the scan rate was 1 mV s⁻¹. (b) Exchange current density and kinetic current density (@50 mV) of NiCuCr/C catalyst with different Cu/Ni ratios.

Revision: Page 10, line 196. "We also tested NiCuCr/C with different Cu and Cr contents, and it demonstrated that the copper content of 5% and Cr content of 2% were the optimal (Supplementary Figure 11 and 12), which is the NiCuCr/C composition used in this paper."

Comment 5. *The Cr₂O₃ species enhance the interfacial water binding, How about the amount of Cr₂O₃ on the HOR performance?*

Response: We thank the reviewer's constructive suggestion. The effect of Cr₂O₃ amount to the HOR activity of NiCuCr/C was studied. The amount of surface Cr₂O₃ was adjusted by changing the precursor ratio between $\text{Cr}(\text{NO}_3)_3 \cdot 9\text{H}_2\text{O}$ and $\text{Ni}(\text{NO}_3)_2 \cdot 6\text{H}_2\text{O}$. As shown in Figure R6a, the HOR activity varies according to different Cr/Ni molar ratios. To quantify the HOR activity of the Ni-based catalysts, we calculated the exchange current density and mass activity at 50 mV, and the results were summarized in Figure R6b. It is suggested small amount of Cr₂O₃ enhanced the HOR activity; however, excessive Cr₂O₃

blocked the active site of NiCu. The optimal Cr/Ni ratio was found to be 0.02.

Figure R6. (a) Polarization curves of NiCuCr/C catalyst with different Cr/Ni ratios in H₂-saturated 0.1 M KOH. The catalyst loading was 0.080 mg_M cm⁻² for catalysts. The rotating speed was 1600 rpm and the scan rate was 1 mV s⁻¹. (b) Exchange current density and kinetic current density (@50 mV) of NiCuCr/C catalyst with different Cr/Ni ratios.

Revision: Page 10, line 196. “We also tested NiCuCr/C with different Cu and Cr contents, and it demonstrated that the copper content of 5% and Cr content of 2% were the optimal (Supplementary Figure 11 and 12), which is the NiCuCr/C composition used in this paper.”

Comment 6. NiCuCr/C shows high performance in HEMFC, how does it compare with commercial Pt/C?

Response: We thank the reviewer for the valuable suggestion. The HEMFCs using Pt/C and PtRu/C as anodes were tested and the polarization and power density curves were shown in Figure R7a. Note, much higher anode metal loading was used for NiCuCr HEMFC, because of its low price. At the region of the current density lower than 700 mA cm⁻², the cell voltage of the NiCuCr HEMFC was only about 40 mV lower than the Pt based HEMFC, demonstrating that by increasing the anode catalyst loading, the performance of the NiCuCr HEMFC could be quite close to the Pt based HEMFCs (Figure R7b). However, when current density was high than 700 mA cm⁻², the NiCuCr HEMFC showed a dramatic decrease of the cell voltage, which was attributed to the mass transfer issue caused by the high anode loading and the deactivation of Ni-based catalysts at high potential. Thus, it suggested that to further improve the peak power density of the Ni-based HEMFC (Figure R7b), we should further increase its activity and the potential stability.

Figure R7. (a) H₂-O₂ HEMFC polarization and power density curves with Pt/C, PtRu/C or NiCuCr/C anode. The anode catalyst loading was 9 mg_{n-PGM} cm⁻² and 0.2 mg_{PGM} cm⁻², respectively. And the cathode catalyst loading was 0.2 mg_{Pt} cm⁻² using commercial 40% Pt/C. The anode, cathode humidifier temperatures, and the cell temperature were 78, 80, and 80 °C, respectively. The back pressure was 2.0 bar, and the flow rates of H₂ and O₂ were both 1000 sccm. (b) The cell voltage at 700 mA cm⁻² and peak power density of NiCuCr/C, Pt/C or PtRu/C, respectively.

Revision: Page 13, line 253. “The NiCuCr/C HEMFC even shows competitive cell performance with Pt-based HEMFC at the current density region below 700 mA cm⁻² (Supplementary Figure 20). However, at high current density region, the NiCuCr/C HEMFC still has clear gap to the Pt-based HEMFCs.”

Reviewer #2:

General Comments. *The key finding in this work is the unique roles that Cu and Cr plays in the NiCuCr nano-electrocatalyst for HOR. The Cu weakens the hydrogen binding while the Cr strengthens the interfacial water binding, both of which enhance the performance of HOR. The findings were used to substantiate the reasons why the catalyst performed better than many previous literature on the same subject of HOR with PGM-free catalysts. The work is well conducted, well written, with all interpretations based on the experimental and theoretical findings and should be considered for publication in the Nature Communications after minor revisions.*

Response: We thank the reviewer for the kind comments of our work. All of your comments are highly important and valuable for us to improve the manuscript. We addressed the comments point-by-point and made the corresponding changes accordingly in the revised manuscript.

Comment 1. *While we understand the reason for Ni, what informed the choice of Cu and Cr in this work? Would other metallic elements in the same periodic table exhibit same properties?*

Response: We thank the reviewer for the valuable question. The calculation and experimental results demonstrated that the Ni has stronger HBE than the optimum, thus weaken the hydrogen binding is in favor of promoting the HOR performance (*Electrochimica Acta* **305**, 452-458 (2019); *Coord Chem Rev* **478**, 214980 (2023)). Cu has weak HBE, and Cu is miscible with Ni. Thus, we chose to alloy Ni with Cu. From the view of apparent HBE, increasing the water binding was in favor of weaken the HBE_{app}. The Cr³⁺ is the one of the ions with lowest enthalpies of hydration (*J Chem Educ* **54**, 540 (1977)), suggesting the strong binding to water. Thus, we chose the NiCuCr combination.

Based on our experimental results, Co, Mo, and W could also optimize the HBE on Ni, and thus promoting their HOR performances. However, Cu and Cr were the best choice and the NiCuCr give the highest HOR performance.

Revision: Page 4, line 70. "Specifically, to optimize the intrinsic HBE, we select Cu as alloying agents to weaken the hydrogen adsorption of Ni, because Cu has weak HBE and is miscible with Ni. To optimize WBE, a more hydrophilic element Cr in form of oxide is decorated on the surface of the catalyst to manipulate the solvation environment. The Cr³⁺ is the one of the ions with lowest enthalpies of hydration, suggesting the strong binding to water⁵⁰."

Comment 2. *Figure 5C, what is the source of the lattice oxygen? It seems everything is attributed to water, why? Could the alloys have some oxide phases? It may not be possible to remove all the oxide by 1 h annealing under 5% H₂/95% Ar.*

Response: Thanks for your valuable concerns and remarks. The reviewer is correct and the lattice oxygen is originated from the oxide phases existed in the sample. The oxides cannot be completely removed, especially our sample was stored in air before the XPS tests. We have done control experiments to confirm the source of the lattice oxygen. Two control samples were obtained. One is the NiCuCr/C stored in air condition for 2 months, and the other is the NiCuCr/C calcined in synthetic air at 470 °C for 2 h. Both ways promote the formation of the oxides. As shown in Figure R9, the XPS results show the increase of the portion of lattice oxygen in the O 1s spectra for both control samples, and the portion of the Ni(0) in Ni 2p spectra was reduced accordingly, confirming the lattice oxygen is originated from oxides. We have revised the manuscript to clarify it.

Figure R9. High resolution (a) O 1s XPS spectra and (b) Ni 2p spectra of NiCuCr/C, NiCuCr/C-2 month storage and NiCuCr/C air calcination.

Revision: Page 18, line 343. “The lattice oxygen was originated from the surface oxides, because the samples were stored in air.”

Comment 3. What about the XPS of the carbon support? Also, why is it not seen in the XRD?

Response: Thanks for your valuable suggestion. The C 1s XPS spectra of the NiCuCr/C was shown in Figure R10a. The peak can be deconvoluted into three peaks at 284.8, 286.1 and 289.5 eV, assigned to the C-C, C-O and C=O bond, respectively (*Angew Chem Int Ed* **58**, 10644-10649 (2019)).

To confirm the XRD peak position of the carbon support, we synthesized the carbon support only (denoted as C) by using same procedure for NiCuCr/C catalyst but excluded all metal precursors. As shown in Figure R10b, the C shows a broad peak at *ca.* 26° in XRD, corresponding to the character peak of Graphite (PDF Card No. 26-1080). The peak is weak, so it is not sharp in the XRD pattern of NiCuCr/C. However, a shoulder could be observed in the XRD pattern of NiCuCr/C at the same location, which was shown in the enlarged inset of Figure R10b.

Figure R10. (a) High resolution C 1s XPS spectra of NiCuCr/C. (b) XRD pattern of C and NiCuCr/C.

Revision: Page 7, line 133. “The C 1s XPS peak of the NiCuCr/C (Supplementary Figure 8) can be deconvoluted into three peaks at 284.6, 286.1 and 290.0 eV, assigned to the C–C, C–O and C=O bond, respectively.”

Page 5, line 96. “The soulder at *ca.* 26° in the XRD pattern was cotributed by the carbon support.”

Comment 4. Check the grammar. For example, p.4 line 55 "significantly influent" is a wrong grammar. please correct.

Response: Thanks for the notification. We have double checked the grammar and revised the manuscript thoroughly.

Revision: Page 4, line 56. “The adsorbed hydroxide is found significantly influencing the HOR kinetics in alkaline, and the bifunctional mechanism is proposed.”

Reviewer #3:

General Comments: In the manuscript by Wang et al., the authors report the NiCuCr/C electrocatalyst for the hydrogen oxidation reaction in alkaline conditions. A high activity was observed for the NiCuCr/C in RDE test, and the MEA using the NiCuCr/C demonstrated high peak power density as well, illustrating the high promising of this reported catalyst. The authors attributed the high performance to the optimized apparent hydrogen binding, which coming from the alloy effects and the surface oxides. Some in-situ characterizations, such as ATR-SERIES was employed to investigate the reaction mechanism.

The presented data are well organized and the manuscript was well written, I would recommend the manuscript for publication in Nature Communications after solving the following issues.

Response: Thanks for your remarks and kind recommendation for our work. All of your comments are highly important and valuable for us to improve the manuscript. We addressed the comments point-by-point and made the corresponding changes accordingly in the revised manuscript.

Comment 1. The Cr_2O_3 cluster on the surface was considered important for the tuning of the apparent hydrogen binding energy. However, the Cr_2O_3 was possible to be dissolved in alkaline condition. I wonder are the Cr_2O_3 survived after HEMFC test? What is the chemical state of Cr after stability test? Please specify in the manuscript.

Response: Thanks for your valuable comments. According to the Pourbaix diagram for Cr-H₂O system (*Corros Sci* **51**, 807-819 (2009), (Figure R11), Cr_2O_3 is stable below pH 14 and below 0.6 V vs. RHE. Thus, in the RDE test using 0.1 M KOH electrolyte, the Cr_2O_3 can be survived.

Figure R11. Pourbaix diagram of Cr-H₂O system at 25°C (Reproduced with permission from *Corros Sci* **51**, 807-819 (2009). Copyright 2023, Elsevier).

For the HEMFC test, we conducted XPS characterization of NiCuCr/C before and after stability test. As shown in Figure R12a, the Ni 2p XPS spectra shows a decrease of Ni⁰ and increase of Ni²⁺, revealing

a growth of hydroxide. For the Cr 2p spectra of the NiCuCr/C before the stability test, two major peaks were observed at 577.5 and 587.3 eV, which were attributed to the Cr³⁺ 2p 1/2 and 2p 3/2 peaks, respectively (Figure R12b). For the NiCuCr/C after the stability test, although the spectra became noisy because of the added ionomer in the MEA test, the two peaks with almost identical locations were observed, indicating the Cr maintained +3 state after the HEMFC test. The Ni/Cu/Cr ratio measured from XPS was 1/0.165/0.081 at initial and 1/0.089/0.080 after the stability test. It demonstrated that most of the Cr survived in the stability test, but partial Cu was leached.

Figure R12. High resolution (a) Ni 2p and (b) Cr 2p spectra of NiCuCr/C and NiCuCr/C after 156 h HEMFC test.

Revision: Page 15, line 279. “The Ni 2p XPS spectra (Supplementary Figure 23a) illustrated increased oxidized Ni portion after the stability test, and the Cr 2p XPS spectra (Supplementary Figure 23b) demonstrated that the Cr maintained as Cr³⁺. The Ni/Cu/Cr ratio measured from XPS was 1/0.165/0.081 at initial and 1/0.089/0.080 after the stability test. It demonstrated that most of the Cr survived in the stability test, but partial Cu was leached.”

Comment 2. In HEMFC stability test (Figure 4c), sudden drops of cell voltage were observed. What was the reason? Did they come from the degradation of the catalysts?

Response: We thank the reviewer for the valuable question. In HEMFC stability test, the sudden drop of current density was attributed to the fluctuated of the reaction temperature, which mainly caused by the water feeding to the humidification tank. Figure R13 shows the anode/cathode temperature curves during NiCuCr/C HEMFC stability test. It clearly demonstrates that each fluctuate of the anode/cathode temperature caused a sudden increase of the high frequency resistance (HFR), leading to the drop of current density. Because the current density was recoverable, we conclude that the catalyst did not degrade.

Figure R13. H₂-O₂ HEMFC real-time conditions (anode/cathode/cell temperature, HFR and current density/cell voltage). The anode NiCuCr/C catalyst loading was 9 mg_{metal} cm⁻². The cathode catalyst loading was 0.2 mg_{Pt} cm⁻² using commercial 40% Pt/C. The anode, cathode humidifier temperatures, and the cell temperature were 78, 80, and 80 °C, respectively. The back pressure was 2.0 bar.

Comment 3. *PtRu/C* was generally used as the benchmark anode catalyst for the anion exchange membrane fuel cells. It would be better if the fuel cell performance of the catalyst is compared with the *PtRu/C*, although the Ni-based catalysts always has worse performance than the PGM catalysts.

Response: We thank the reviewer for the valuable suggestion. The HEMFCs using Pt/C and PtRu/C as anodes were tested and the polarization and power density curves were shown in Figure R14a. Note, much higher anode metal loading was used for NiCuCr HEMFC, because of its low price. At the region of the current density lower than 700 mA cm⁻², the cell voltage of the NiCuCr HEMFC was only about 40 mV lower than the Pt based HEMFC, demonstrating that by increasing the anode catalyst loading, the performance of the NiCuCr HEMFC could be quite close to the Pt based HEMFCs (Figure R14b). However, when current density was high than 700 mA cm⁻², the NiCuCr HEMFC showed a dramatic decrease of the cell voltage, which was attributed to the mass transfer issue caused by the high anode loading and the deactivation of Ni-based catalysts at high potential. Thus, it suggested that to further improve the peak power density of the Ni-based HEMFC (Figure R14b), we should further increase its activity and the potential stability.

Figure R14. (a) H₂-O₂ HEMFC polarization and power density curves with Pt/C, PtRu/C or NiCuCr/C anode. The anode catalyst loading was 9 mg_{n-PGM} cm⁻² and 0.2 mg_{PGM} cm⁻², respectively. And the cathode catalyst loading was 0.2 mg_{Pt} cm⁻² using commercial 40% Pt/C. The anode, cathode humidifier temperatures, and the cell temperature were 78, 80, and 80 °C, respectively. The back pressure was 2.0 bar, and the flow rates of H₂ and O₂ were both 1000 scfm. (b) The cell voltage at 700 mA cm⁻² and peak power density of NiCuCr/C, Pt/C or PtRu/C HEMFCs, respectively.

Revision: Page 13, line 253. “The NiCuCr/C HEMFC even shows competitive cell performance with Pt-based HEMFC at the current density region below 700 mA cm⁻² (Supplementary Figure 20). However, at high current density region, the NiCuCr/C HEMFC still has clear gap to the Pt-based HEMFCs.”

Comment 4. The study regarding stability for Ni based catalyst (*ACS Appl Mat Interfaces* 12, 31575-31581 (2020)) should be cited and added the corresponding data in Figure 4b.

Response: We thank the reviewer for pointing this out. We have included the corresponding data and cited the literature, as shown in Figure R15.

Figure R15. Summary of the peak power density and durability of the reported PGM-free anode MEA.

Revision: We revised Figure 4b and included the Ni@C-500 catalyst reported in ACS Appl Mat Interfaces 12, 31575-31581 (2020).

Comment 5. *The Ag/C was used as the cathode to assemble non-Pt MEA. Why Ag/C was selected as one of the precious metals? Please add more explanation in the manuscript.*

Response: We thank the reviewer for the valuable question. Ag has been reported as an efficient ORR catalyst in the last decade (*Nat Chem* **6**, 828-834 (2014); *ACS Appl Energy Mater* **1**, 1990-1999 (2018); *ChemElectroChem* **6**, 73-86 (2019); *Chem Eng J* **446**, 136966 (2022)), and it has been used as the cathode for HEMFC (*Proc Nat Acad Sci USA* **105**, 20611-20614 (2008); *J Mater Chem A* **6**, 15404-15412 (2018); *Int J Hydrogen Energy* **48**, 11058-11070 (2023)). Although Ag belongs to precious metal, its price (~\$0.85 g⁻¹) is much lower than Pt (~\$32.8 g⁻¹). In our experiment, the Ag loading was only 0.6 mg_{Ag} cm⁻², and the fabricated HEMFC delivered a peak power density of 335 mW cm⁻². Thus, the cost for the Ag cathode is only \$1.4 kW⁻¹. The US DOE set the ultimate goal of fuel cell cost is \$30 kW⁻¹ (*MRS Bull* **45**, 57-64 (2020)). Thus, the Ag based cathode catalyst cost would not be the major part for the fuel cell cost.

Revision: Page 15, line 287. “Ag showed high ORR performance in alkaline condition, but much lower cost than PGMs. And high performance HEMFCs using Ag as the cathode have been reported.”

Comment 6. *Some experimental details are missing, for example, the test temperature for the LSV and chronoamperometry curve shown in Figure 3, the MEA test condition for the stability shown in Figure 3c.*

Response: We thank the reviewer for pointing this out. We have added the experiments details in the caption of Figure 3a, and have revised the MEA test conditions in the Supplementary Information.

Revision:

Page 14, the caption of Figure 3, line 208, “Polarization curves of Ni/C, NiCu/C, NiCr/C, NiCuCr/C and commercial Pt/C in H₂-saturated 0.1 M KOH at 25 °C.” line 215, “Chronoamperometry test of NiCuCr/C at 0.09 V in pure H₂ or H₂ containing 100 ppm CO-saturated electrolyte at 25 °C.”

Page 15, the caption of Figure 4, line 266. “H₂-O₂ HEMFC stability test curve of NiCuCr/C and Ni/C at 0.7 V, the test conditions were same as polarization test except the flow rate of H₂ and O₂ were both 500 sccm.”

Page 21, line 415. “All electrochemical measurements were conducted at 25 °C.”

REVIEWERS' COMMENTS

Reviewer #1 (Remarks to the Author):

The authors addressed all my comments. It can be considered as a publication in the present form.

Reviewer #3 (Remarks to the Author):

All the comments have been carefully addressed, therefore I recommend that it be accepted now.